# Current Advancements and Future Perspectives of Immunotherapy in Breast Cancer Treatment

**Maria Vasileiou** [1],*[ID]**, Savvas Papageorgiou** [2][ID] **and Nam P. Nguyen** [3][ID]

1   Department of Pharmacy, School of Health Sciences, National and Kapodistrian University of Athens, 15771 Athens, Greece
2   Department of Molecular and Cell Biology, University of Leicester, Lancaster Road, Leicester LE1 7RH, UK
3   Department of Radiation Oncology, Howard University, Washington, DC 20060, USA
*   Correspondence: mariavasileiou65@gmail.com

**Abstract:** Breast cancer is the most commonly diagnosed cancer in women and is a leading cause of cancer death in women worldwide. Despite the available treatment options, such as surgery, chemotherapy, radiotherapy, endocrine therapy and molecular targeted therapy, breast cancer treatment remains a challenge. The advent of immunotherapy has revolutionized the treatment of breast cancer as it utilizes the host's immune system to directly target tumor cells. In this literature review, we aim to summarize the recent advancements made in using immunotherapy for treating breast cancer patients. We discuss the different types of existing immunotherapies for breast cancer, including targeted therapy using monoclonal antibodies against breast cancer specific antigens and the use of immune checkpoint inhibitors to elicit an immune response against cancer cells. Finally, we consider the development of breast cancer vaccines that train the immune system to specifically recognize cancer cells and the future perspectives of immunotherapy for breast cancer.

**Keywords:** breast cancer; immunotherapy; monoclonal antibodies; immune checkpoint inhibitors; vaccines; cytokines; CAR-T cells

## 1. Introduction

Immunotherapy refers to harnessing the patient's own immune system to eradicate cancer. The first immunotherapy attempt was introduced in the late 19th century by two German physicians, Fehleisen and Busch, who noticed significant tumor regression after erysipelas infection [1]. The following years, the role of immunotherapy was extensively studied by William B. Coley, also known as the "Father of Immunotherapy", who conducted experiments which involved injections of live bacteria *S. pyogenes* and *S. marcescens* (referred to as Coley's toxin) into patients with inoperable bone cancers. In recent years, accumulating data support a key role for the immune system in determining both the response to standard and adjuvant therapy in patients with breast cancer [2].

The immune system consists of monitoring mechanisms that detect and respond to the presence of cancer cells, a process called "immunosurveillance" [3,4]. Specifically, immunosurveillance consists of three phases: elimination, equilibrium and escape [5]. During the elimination phase, the immune system cells recognize tumor-specific antigens and respond by destroying tumor cells. The latter is achieved due to tumor immunogenicity as well as the interaction of tumor-specific antigens with immune system cells. Immunogenicity refers to the ability of a tumor to induce immune responses which inhibit its survival and proliferation. Next, tumor cells that survive the elimination phase, enter a state of equilibrium, which is a transition period from the elimination phase to the onset of the disease [6]. Lastly, tumor cells manage to escape immune surveillance and proliferate after establishing an immune suppressive tumor microenvironment (TME) [7]. The acquired knowledge of immune surveillance and immunogenicity has revolutionized the immunotherapy treatment landscape of several types of cancers, including breast cancer.

Breast cancer (BC) is the most commonly diagnosed cancer among women worldwide with 2.26 million new cases in 2020 and 684,996 deaths [8]. According to the World Health Organization, malignancies account for 107.8 million disability-adjusted life years (DALYs), of which 19.6 million DALYs are attributed to breast cancer [9]. The significant prognostic differences in patient outcomes have led to the classification of the disease based on estrogen receptor (ER) and progesterone receptor (PR) status, as well as the expression of human epidermal growth factor receptor 2 (HER2) into luminal A (defined as ER+ and/or PR+, HER2−), luminal B (defined as ER+ and/or PR+, HER2+), HER2 overexpressing (defined as EGFR+, ER−, PR−) and triple negative breast cancer (TNBC) which is defined by the absence of ER, PR and HER2. TNBCs represent 10–20% of all BCs with 71–91% of them having a basal-like phenotype and approximately 20% expressing ER or HER2 to a certain extent [10,11]. TNBC is diagnosed based on immunohistochemistry (IHC) in clinical practice. However, guidelines recommend additional analysis in IHC samples with ambiguous HER2 status in order to avoid false positive or negative results [12].

The number of risk factors of breast cancer is significant and includes both modifiable factors and non-modifiable factors. Non-modifiable factors include age, sex, ethnicity, genetic mutations, family and reproductive history and previous radiation therapy, while modifiable factors include body mass index (BMI), physical activity, alcohol and processed food intake, smoking, exposure to certain chemicals, vitamin supplementation and chosen drugs. It is estimated that about 80% of patients with BC are white non-Hispanic female individuals aged >50 with mutations in the *BRCA1*, *BRCA2*, *TP53*, *CDH1*, *PTEN* and *STK11* genes and less often in the *ATM*, *PALB2*, *BRIP1*, *CHEK2* and *XRCC2* genes. In addition, they present with an increased BMI, long smoking history, insufficient vitamin D supplementation, as well as excessive consumption of alcohol and highly processed meat, which is classified as a Group 1 carcinogen [13].

Currently, the conventional therapeutic approach for BC involves surgery, chemotherapy, radiotherapy, endocrinotherapy and molecular targeted therapy. Due to the potential of immunotherapy, among various chemotherapy agents, it has drawn the attention of researchers. Clinical evidence shows significant variability in the treatment response to programmed cell death-ligand 1 (PD-L1) inhibitors versus standard chemotherapy agents. According to the randomized, phase 3 KEYNOTE-119 study (NCT02555657), there was no significant difference observed in overall survival (OS); however, the results suggested a potential positive association between tumor mutational burden and pembrolizumab versus single-agent chemo per investigator's choice of capecitabine, eribulin, gemcitabine or vinorelbine [14]. Similarly, positive results were observed in the phase III IMpassion130 trial comparing atezolizumab plus chemotherapy versus chemotherapy plus placebo. In fact, there was an increase in overall survival outcome only in PD-L1+ TNBC patients [15]. The American Society for Radiation Oncology, the Society for Immunotherapy of Cancer, and the US National Cancer Institute have investigated the incorporation of immunotherapy into radiotherapy [16]. This initiative is supported by a single-arm clinical trial of 22 patients with metastatic melanoma, followed by the administration of ipilimumab, which led to a partial reduction in the sizes of non-irradiated metastatic sites in four (18%) patients [17]. Thus, the benefit of immunotherapy versus or combined with conventional treatments is evident and there is ongoing investigation.

This literature review elaborates on various immunotherapies currently being applied for breast cancer treatment. These include 1. monoclonal antibodies; 2. immune checkpoint inhibitors: an approach designed to 'unleash' T cell responses; 3. breast cancer vaccines which train the immune system to fight BC; 4. cytokine therapies: an immunomodulatory approach; and 5. adoptive cellular therapies which are based on delivering engineered immune cells into the body to fight cancer. We also discuss preclinical and clinical advancements of various immunotherapies, limitations, and future directions.

## 2. Monoclonal Antibodies against HER-2 Receptor Protein

According to the National Institute of Health, monoclonal antibodies are artificially made proteins that can recognize specific targets and are widely used as a targeted cancer therapy [18]. Most often, they are categorized into those targeting immune molecules, otherwise known as immune checkpoint inhibitors, or into those targeting oncogenic membrane receptors [19]. Figure 1 demonstrates the two types of monoclonal antibodies that will be discussed in this review.

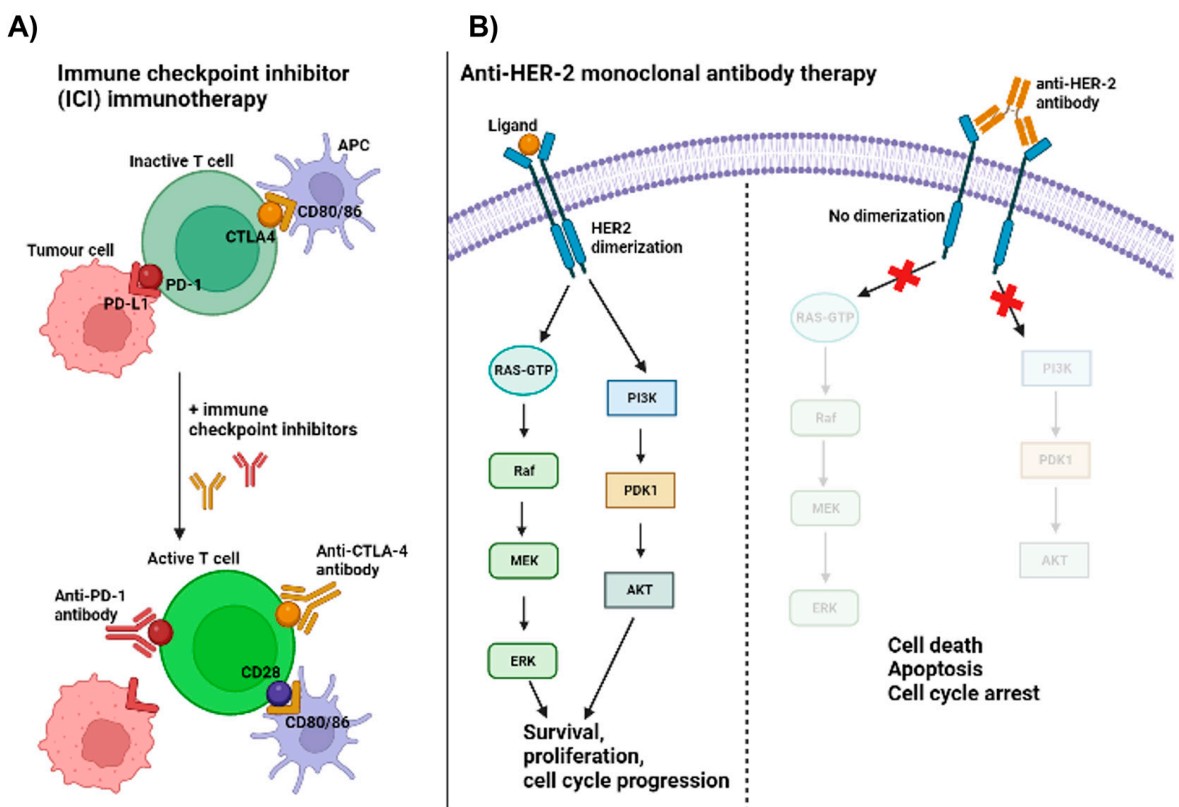

**Figure 1.** Monoclonal antibody immunotherapies. (**A**) Demonstrates the mechanism of action of anti-PD-1 and anti-CTLA-4 immune checkpoint inhibitors. T cells require a total of 3 signals for proper activation. Antigen-specific recognition through T cell receptor (TCR): major histocompatibility complex (MHC) interaction between costimulatory molecules and cytokine signals [20]. Cytotoxic T-lymphocyte–associated antigen 4 (CTLA-4) and programmed cell death-1 (PD-1) are negative checkpoint molecules that drive the inactivation of T cells upon interaction with their corresponding ligands, CD80/86 on antigen-presenting cells (APCs) and PD-L1 on tumor cells. Immune checkpoint inhibitors block that interaction and allow positive checkpoint molecules (e.g., CD28) to interact with their corresponding ligands (i.e., CD80/86) and deliver the 2nd signal for T cell activation. (**B**) Illustrates the mechanism of action of an anti-HER-2 monoclonal antibody by preventing dimerization of HER subunits. This ultimately leads to the perturbation of downstream signaling events and consequently promotes cell death, apoptosis and cell cycle arrest (created using BioRender.com, URL accessed on 16 May 2023).

Approximately one third of breast cancer patients exhibit upregulation of HER2 receptor which has led to the development of monoclonal antibodies against this particular receptor such as trastuzumab or pertuzumab [21,22]. Trastuzumab, commercially known as Herceptin, was the first humanized monoclonal antibody to be approved by the Food and Drug Administration (FDA) in 1998 for metastatic HER2+ breast cancer. It is known to function through multiple mechanisms, from mediating HER2 receptor internalization and subsequent degradation via recruitment of ubiquitin ligase to dampen HER-2-dependent signaling pathways such as the RAS/MAPK or PI3K/Akt pathways [23]. Patients often de-

velop resistance to trastuzumab despite initial responses which has led to the development of alternative anti-HER2 monoclonal antibodies such as pertuzumab [21,24,25].

Pertuzumab is another humanized monoclonal antibody against a different HER2 extracellular domain (domain II) compared to trastuzumab (domain IV). Pertuzumab prevents the heterodimerization of HER receptors which leads to the inactivation of downstream signaling pathways such as PI3K/Akt [26–29]. If having to compare the two antibodies, pertuzumab has shown better outcomes in the clinic including a better safety profile and a higher efficacy compared to trastuzumab [21].

There are multiple clinical trials either underway or already completed that look into the effects of a dual HER2 blockade using a trastuzumab plus pertuzumab combination therapy. For patients with previously refractory disease following treatment with trastuzumab, Portera et al., (2008) reported cardiac toxicity in more than 50% of participants following administration of the dual antibody therapy which led to the study being brought to an end [30]. The Baselga study exhibited similar objective response rates (ORRs) to the Portera study (18% versus 24.5%) but reported no cardiac symptoms [30,31]. On the other hand, the CLEOPATRA study is looking into the effects of a dual HER2 blockade in metastatic patients with no prior treatment. So far, evidence suggests that the combination of the two antibodies as a first line treatment in metastatic breast cancer significantly augments the progression-free survival (PFS) rates [32].

Antibody-drug conjugates (ADCs) are made up of antibodies fused together to cytotoxic agents that allows the efficient delivery of these agents within cancer cells. Antibodies allow for specific recognition of malignant cells followed by ADC internalization and the release of cytotoxic effects mediated by the drug within the cell [33]. Trastuzumab emtansine (T-DM1) is an example of an ADC that is composed of trastuzumab as the monoclonal antibody linked to DM1, a cytotoxic agent that targets microtubule assembly and consequently leading to cell death [34]. In vitro studies have reported remarkable sensitivity to T-DM1 with various different breast cancer cell lines which was also replicated in in vivo animal models [34,35]. There are reports of multiple clinical trials having observed a much better efficacy and safety profile in patients with metastatic breast cancer with T-DM1 compared to trastuzumab or other therapies. Nonetheless, most patients do eventually acquire resistance to T-DM1 [36–38].

This has led to the development of alternative ADCs such as Trastuzumab deruxtecan. The latter contains the same antibody as T-DM1 but with a topoisomerase I inhibitor instead and was first approved in the USA in 2019 [39]. Using patients previously treated with T-DM1 and having progressed with it, Trastuzumab deruxtecan showed significant clinical efficacy; however, some noticeable adverse effects related to lung disease were observed in a substantial percentage of those patients [40]. Another study compared the efficacy of the two trastuzumab ADCs in patients previously treated with traszutumab plus a taxane. Cortes et al. (2022) reported a lower risk of death or disease progression in patients treated with trastuzumab deruxtecan than those treated with T-DM1. However, the same pulmonary complications observed by Verma et al. (2020) were also reported here [41].

Worth mentioning is also Sacituzumab govitecan, an ADC directed at trophoblastic cell-surface antigen-2 (Trop-2) expressed in multiple solid tumors including breast cancer. The fact that Trop-2 is highly restricted to tumor cells limits off-target effects and spares healthy tissue and makes it the perfect candidate to be used in cancer therapy [42–44]. The drug conjugate used for this ADC is SN-38, another topoisomerase I inhibitor and the ADC as a whole was recently approved in the USA in 2020 for patients with metastatic triple negative breast cancer previously treated and progressed with at least two other therapies [45].

## 3. Immune Checkpoint Inhibitors (ICIs)

Immune checkpoint inhibitors (ICIs) are monoclonal antibodies that specifically target immune checkpoint molecules on the surface of T cells or tumor cells to promote the activation of immune cells and consequently elicit an immune response against the cancer

cells [46,47]. Several studies have detected significant levels of the immunosuppressive PD-L1 checkpoint molecule in HER-2+ breast cancer [48–51] which would suggest that PD-L1 blockade with ICIs would be beneficial.

Several clinical studies are evaluating the use of such antibodies in patients with TNBC—a subtype negative for progesterone, estrogen and HER-2 receptors [52,53]. Atezolizumab is an anti-PD-L1 monoclonal antibody that managed to increase the ORR in patients with metastatic TNBC by approximately 20% compared to non-treated patients [52]. Other PD-L1 checkpoint inhibitors such as avelumab have also been shown to augment the ORR in TNBC patients [53]. In both cases, the efficacy of the antibodies seemed to be dictated by the level of PD-L1 expression in patients. Those with higher expression seemed to benefit more compared to those with lower to no expression [52,53]. Due to the ability of chemotherapy agents to promote antigen presentation and PD-L1 expression, several studies have turned their attention to combining these agents with ICIs [54,55]. The anti-PD-L1 monoclonal antibody, pembrolizumab, was combined with nab-paclitaxel, paclitaxel and gemcitabine/carboplatin chemotherapy in metastatic TNBC patients and led to augmented PFS [54]. This again was restricted to PD-L1-positive patients. Furthermore, the effects of nivolumab combination with different chemotherapy agents again in patients with metastatic TNBC was assessed in the phase 2 TONIC trial. The administration of nivolumab following chemotherapy with doxorubicin rather than cisplatin seemed to be more effective in enhancing the ORRs of these patients—35% versus 23% in doxorubicin versus cisplatin-treated patients, respectively [55].

The KEYNOTE trial was the first clinical trial ever to use an anti-PD-1 agent, pembrolizumab, in patients diagnosed with PD-L1+ metastatic TNBC. Overall, the different trial phases suggest that pembrolizumab exhibits a low toxicity profile; however, response rates including ORR and OS remained relatively low. Furthermore, the study failed to show that pembrolizumab was superior to chemotherapy as a monotherapy treatment for metastatic TNBC patients [56–58]. The PANACEA trial enrolled patients diagnosed with HER2+ breast cancer previously progressed on trastuzumab monotherapy to assess the effect of pembrolizumab plus trastuzumab combination therapy. Overall, 15% of PD-L1+ patients had an objective response as opposed to none of the PD-L1- patients in the study. Moreover, no dose-limiting toxicities were reported, and fatigue was the most common treatment-related side effect observed in patients [59].

There are several other clinical trials either underway or already completed that assess the efficacy of different checkpoint inhibitors. Table 1 lists some of those trials.

**Table 1.** List of clinical trials for immune checkpoint inhibitors.

| Therapy Evaluated | Trial Phase | Breast Cancer Setting | Sample Size | Results |
|---|---|---|---|---|
| JS001 [60] | Phase I | TNBC | 20 | Favorable safety profile<br>Moderate response |
| Nivolumab plus radiotherapy [61] | Phase I/II | HER-2 negative metastatic | 28 | Favorable safety profile<br>Modest anti-tumor response |
| Pembrolizumab plus T-DM1 [62] | Phase Ib | HER-2 positive metastatic | 20 | Favorable safety and tolerability profile |
| Ipilimumab plus Nivolumab [63] | Phase II | Unresectable or metastatic metaplastic | 17 | Response up to 3 years<br>Variable responses in patients<br>No new adverse effects reported |
| Pembrolizumab plus gemcitabine [64] | Phase II | HER-2 negative advanced cancer | 36 | Variable responses<br>Tumor landscape mapping warranted for this treatment |
| Nivolumab plus paclitaxel and bevacizumab [65] | Phase II | EGFR-2-negative metastatic | 57 | Favorable efficacy and tolerability |

Table 1 lists some of the clinical trials either underway or completed that have evaluated efficacy of different immune checkpoint inhibitor monotherapies or combined with other modalities such as chemotherapy or radiotherapy. The trial phase, subtype of breast cancer and sample size are also provided for each trial.

## 4. Breast Cancer Vaccines

Cancer vaccines is another type of immunotherapy that uses tumorigenic antigens as a way to train the immune system to recognize those antigens and initiate an immune response against them [66]. There are several types of cancer vaccines including those who utilize a patient's own cancer antigens in order for the vaccine to be specific for the patient's tumor or those who make use of a patient's own immune or tumor cells [67]. Protein-based vaccines are considered to be the conventional method of vaccination and Figure 2 briefly summarizes how they function.

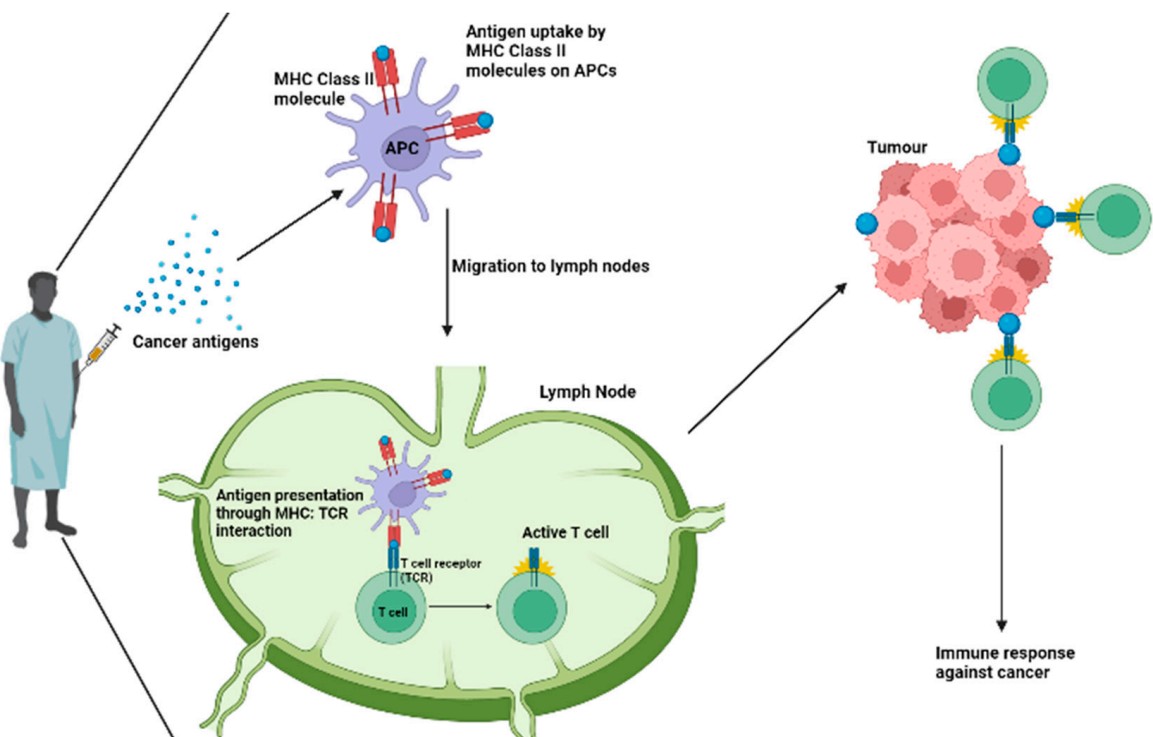

**Figure 2.** Mechanism of action of protein-based cancer vaccines. Protein-based cancer vaccines are used to train T cells to recognize specific tumorigenic antigens. Cancer antigens are taken up by antigen presenting cells (APCs) such as dendritic cells through MHC Class II molecules. Active APCs migrate to lymph nodes and present antigens to T cells through MHC: TCR interactions which lead to T cell activation. Active T cells are then able to recognize those same antigens on the surface of cancer cells and initiate an immune response against those tumor cells (created using BioRender.com, URL accessed on 16 May 2023).

As of 2021, there are several clinical trials underway that evaluate the effectiveness of cancer vaccines in a breast cancer setting. The majority of those trials make use of peptide-based vaccines; however, some of them assess cell-based vaccines as well [68,69].

HER-2 positive breast cancer has been reported as a subtype with a strong association between its tumor immune landscape and disease progression [70]. More specifically, the NeoALTTO trial has observed a reduction in disease progression in patients with this particular breast cancer subtype following tumorigenic T cell infiltration [71]. Moreover, studies have also provided evidence that immune responses against HER-2 positive tumors are initiated by both CD8+ and CD4+ T lymphocytes [72]. The evidence for the link between disease recurrence and immune cell infiltration for this cancer subtype is overwhelming; thus, several clinical trials have been targeting this receptor using either protein-based or cell-based vaccination. For example, some of these make use of anti-HER-2/3 vaccines, dendritic cells, autologous or allogenic tumor cells and anti-Arg1 vaccines [69].

The field of DNA vaccine development is rapidly evolving, especially having gone through the COVID-19 pandemic. The AST-301 (pNGVL3-hICD) is a DNA-based vaccine that contains the pNGVL3-hICD plasmid which encodes the intracellular domain of HER-2 receptor protein [73]. A clinical trial evaluating the clinical efficacy of the following vaccine plus GM-CSF therapy is still underway. Results so far have shown that immunization with the following vaccine promotes the activation of anti-HER-2 T lymphocytes [74,75]. In vivo studies with HER-2+ gastric cancer models have shown a favorable safety profile and significant activation of anti-tumor immune responses and consequently tumor growth regression in mice treated with the vaccine [73].

Other than HER-2 specific vaccines, other tumorigenic antigens are also being explored as candidates for vaccine development. For example, Globo H is a carbohydrate antigen found on tumor cells including breast cancer [76]. Adagloxad simolenin (OBI-822) is a protein-based vaccine that encompasses the Globo H antigen covalently linked to an immunogenic carrier protein known as keyhole limpet hemocyanin (KLH). The latter allows for the vaccine to be delivered in close proximity to T cells situated specifically near the targeted protein—in this case Globo H. A phase II trial made use of the following vaccine in patients diagnosed with metastatic breast cancer and has demonstrated enhanced immune responses against Globo H-positive cells and an overall promising tolerability profile [77].

## 5. Cytokine Therapies

Cytokines consist of small proteins that are released upon specific stimuli to promote numerous cellular processes. Cytokines can have both anti- and pro-inflammatory effects and can promote cell growth or death [78]. For example, interleukin-1β (IL-1β) is a cytokine often found within the TME and can have both anti- and pro-tumorigenic effects. In vivo studies using breast cancer models reported that IL-1β deficiency results in tumor regression due to the intratumoral recruitment and differentiation of inflammatory monocytes. IL-1β deficiency caused low levels of macrophages and recruitment of CD11b+ dendritic cells which consequently led to an interleukin-12 (IL-12) upregulation and interleukin-10 (IL-10) downregulation. This promoted anti-tumor immunity and CD8+ lymphocyte activation. For that reason, the rationale of combining anti-IL-1β and anti-PD-1 ICI treatments was suggested which led to tumor regression. Blocking IL-1β not only reduced tumor progression but also facilitated checkpoint inhibition [79].

Interleukin-2 (IL-2) is a cytokine often used in cancer immunotherapy mostly due to its ability to activate and expand immune cells, including T lymphocytes and natural killer (NKs) cells. For that reason, studies are looking into IL-2 intervention as a treatment course for cancer in general. For example, a recently commenced clinical trial is aiming to recruit TNBC patients to assess the efficacy of IL-2 administration [80]. In addition, the combination of IL-2 with chemotherapy has also been proven to improve the response rate and survival of patients with advanced breast cancer [78]. TNBC patients are often reported to be positive for epidermal growth factor receptor (EGFR) which has given enough reason for anti-EGFR antibodies such as cetuximab to be considered as a treatment option [81]. Roberti et al. (2011) have demonstrated that administration of either IL-1 or IL-15 can enhance the cetuximab-mediated antibody dependent cell cytotoxicity (ADCC) by NKs in different EGFR-positive TNBC cell lines. Although both cytokines doubled the lytic activity of NKs, IL-15 showed slightly superior results compared to IL-2 [81].

Interleukin-12 (IL-12) is a cytokine that plays a crucial role in the immune response against cancer. Similar to IL-2, it has been shown to activate T cells, NKs and dendritic cells which together can lead to tumor regression. For that reason, several studies have explored the approach of using this cytokine as another immunotherapy. For example, an in vitro study using SCK murine mammary carcinoma cells, described that IL-12 and IL-18 synergistically inhibit tumor angiogenesis and tumor development [82]. Furthermore, clinical studies have also demonstrated a favorable safety profile and high efficacy of IL-12 therapy in patients diagnosed with metastatic breast cancer [83]. In addition, the study

conducted by Telli et al. (2021) investigated the effects of intratumoral injection of plasmid IL-12 on CD8+ T cells and *C-X-C motif chemokine receptor 3* (*CXCR3*) gene expression in triple-negative breast tumors. The treatment induced a *CXCR3* gene signature in the tumors associated with an upregulation of checkpoint molecules such as PD-1/PD-L1 and CD8+ T cell recruitment. Consequently, this sensitized patients to anti-PD-1 therapy, resulting in improved clinical outcomes. This study highlights the potential of IL-12 administration as a therapeutic adjuvant option together with checkpoint inhibition for triple-negative breast tumors as well as the importance of the *CXCR3* gene signature as a predictive biomarker for anti-PD-1 therapy [84]. Cytokine therapy has also been shown to be a promising adjuvant for other conventional treatments such as irradiation. For example, Formenti et al. (2018) investigated the effects of focal irradiation and systemic transforming growth factor-β (TGF-β) blockade in metastatic breast cancer. The latter combination resulted in a decrease in the number of circulating tumor cells and an increase in T cell-mediated tumor cell killing. Furthermore, the combination therapy was well-tolerated by patients and resulted in improved overall survival. The study highlights the potential of combining focal irradiation with systemic TGF-β blockade as a promising therapeutic strategy for metastatic breast cancer [85].

Studies have also evaluated the potential of several other interleukins as immunotherapies in breast cancer. For instance, interleukin-15 (IL-15) is a cytokine that shares many similarities with IL-2 and has been shown to have potent anti-tumor effects. A preclinical study using a mouse model of breast cancer found that IL-15 therapy increased the number and activity of NKs and T cells, resulting in the inhibition of tumor growth and metastasis [86]. A phase I clinical trial has also demonstrated the safety and potential efficacy of IL-15 therapy in patients with metastatic breast cancer [87]. Moreover, interleukin-21 (IL-21) is similar to IL-12 in terms of immune cell activation. Preclinical studies using breast cancer models have found that IL-21 therapy inhibits tumor growth and metastasis by increasing the activity of NKs and T cells [88]. A phase I clinical trial also demonstrated the safety and potential efficacy of IL-21 therapy in patients with metastatic breast cancer [89].

Tumor necrosis factor-alpha (TNF-α) is another type of cytokine and there is a lot of debate in terms of its potential in cancer therapeutics mostly because it has been shown to exert both pro- and anti-tumorigenic effects. For example, studies have reported TNFα to recruit immunosuppressive immune cells to the tumor microenvironment whilst other have shown that similar to other ILs, TNFα can activate the immune system as well [90]. However, its use in breast cancer treatment has been limited by its toxic side effects. For that reason, a modified form of TNF-α called PEGylated TNF-α (PEG-TNF-α) has been developed that has a longer half-life and reduced toxicity. Clinical trials have shown that PEG-TNF-α can have antitumor effects in breast cancer, but further studies are needed to determine its optimal use [91]. In vitro studies of TNF-α in TNBC models have illustrated that the expression of TNF-α promotes cellular proliferation while inhibiting apoptotic pathways. When TNF-α was knocked down, cell proliferation was halted, and apoptosis was induced [92]. In vivo models of TNBC have reported that the administration of TNF-α prior to doxorubicin treatment can enhance tumor accumulation of doxorubicin which generates the rationale of testing this combination in patients [93].

Chemokines are small cytokines often found to be involved in cancer progression, Particularly, the C-C chemokine motif ligand 2/C-C chemokine receptor 2 (CCL2/CCR2) pathway has been shown to drive a pro-inflammatory microenvironment which favors tumor progression [94]. The following study evaluated the safety profile and potential efficacy of a CCL2 inhibitor, propagermanium (PG), as an antimetastatic drug in perioperative patients with primary breast cancer. Overall, a low-grade adverse event profile was reported. IL-6 levels were reduced in a dose-dependent manner which, given the fact that IL-6 is implicated in the metastatic dissemination of breast cancer [95], suggests that PG may have the ability to hinder metastasis in human BC. Furthermore, serum CCL2 levels increased and remained high for at least 2 months post-surgery, possibly promoting cancer metastasis. PG might be preferable for CCL2 inhibition compared to anti-CCL2

neutralizing antibodies in human BC. Finally, the study provided a rationale for the efficacy of PG for the inhibition of metastasis in BC patients based on the FBXW7/CCL2 axis, which is important for BC progression. However, further studies will be required to clarify the mechanism of the downregulation of FBXW7 mRNA levels in some BC patients [96].

Cytokine therapies have shown promising results in the treatment of breast cancer. Multiple ILs have been shown to mediate potent anti-tumor effects through immune cell activation and inhibition of tumor growth and metastasis [96,97]. Further clinical trials are needed to determine the safety and efficacy of these cytokine therapies in the treatment of breast cancer.

## 6. Adoptive Cell Therapies (ACTs)

Despite the vast availability of cancer immunotherapies, there is still a substantial proportion of patients who become resistant to those therapies and exhibit disease progression with time. For that reason, adoptive cellular therapies (ACT) which involve manipulation of either intratumoral or peripheral blood immune cells can serve as another option for these patients [98,99]. There are three types of ACTs including tumor-infiltrating lymphocytes (TILs), TCR—modified lymphocytes and chimeric antigen receptor (CAR) T cells [100,101].

### 6.1. Tumor-Infiltrating Lymphocyte (TIL) Therapy

TIL therapy often involves the ex vivo expansion of either CD8+ or CD4+ T cells and their infusion back into the patient [100]. Both allogeneic and autologous transplants have been attempted for breast cancer patients in the past with each approach having its own harms and benefits. For example, allogeneic transplants exhibited significant toxicity whereas autologous transplants were less efficacious compared to allogeneic [102–105]. TIL therapy has been found to be most effective in patients with a high somatic mutational burden as it allows for the expansion of TILs specifically against tumor-specific neoantigens. For example, a case study for a 49-year-old patient diagnosed with ER-positive metastatic breast cancer reported disease regression following a combination of TIL therapy, specifically against proteins mutated in that patient, IL-2 cytokine therapy and PD-1 immune checkpoint inhibition [106]. A similar study identified 72 mutations in a patient with metastatic TNBC generated TILs against that patient's own mutated genes; however, results of this study were not published [107,108]. The presence of TILs versus immunosuppressive cells such as tumor-associated macrophages (TAMs) dictate the prognosis of TNBC patients [109–111]. Those with higher TAM content are more likely to progress and for their tumors to recur compared to those with greater TIL accumulation [109]. Furthermore, those with greater TIL accumulation are also shown to be more sensitive to chemotherapy treatment [110]; hence, one could perhaps suggest the potential combination of TIL therapy and chemotherapy. There are several clinical trials either underway or completed that make use of TIL therapy either alone or in conjunction with another immunotherapy such as ICIs or cytokine therapy that study the effectiveness of given therapies in patients diagnosed with various breast cancer subtypes [112].

### 6.2. Engineered TCR Therapy

Engineered TCR therapy describes the genetic engineering of the TCR of a patient's own T cells to specifically recognize tumor antigens [113]. Most αβTCR-engineered cells can only be used against antigens presented by MHC molecules expressed on immune cells such as dendritic cells [99]. For instance, Qiongshu et al. (2018) successfully engineered CD8+ T cells to specifically recognize PLAC1 in different breast cancer cell lines. PLAC1 is an example of an antigen known to be often expressed in breast cancer [114]. The study reported robust activation of these cells, evident by the enhanced release of interferon-γ (IFNγ) and TNF-α cytokines. Moreover, an increased number of lysed cells were observed as a result of CD8+ T cell activation in PLAC1-positive cell lines as opposed to either PLAC1-negative or PLAC1-silenced cells. Most importantly, the study reported a significantly decelerated tumor growth in vivo following administration of these CD8+ T cells [115].

$\gamma\delta$T engineered cells are more potent compared to the conventional $\alpha\beta$TCR-engineered cells. This is mainly because they express both $\gamma\delta$TCRs and killer cell immunoglobulin-like receptors (KIRs) [116,117]. Several studies have reported the potential of these types of therapy in breast cancer as either a monotherapy [118] or combined with trastuzumab [119]. Regardless, this type of therapy has also been shown to convey immunosuppressive effects on several other immune cells in breast cancer, suggesting that the inhibition of these cells instead could confer anti-tumorigenic effects [120]. For that reason, additional research is warranted as to their potential use in immunotherapeutics. Overall, numerous clinical trials are underway to study the effectiveness of different engineered-TCR therapies in breast cancer against already established cancer targets such as NYESO-1 or MAGE-A3 or other neoantigens [112].

*6.3. Chimeric Antigen Receptor (CAR) T Cell Therapy*

Chimeric antigen receptor (CAR) therapy includes the genetic modification of a patient's own T cells that involves the addition of a CAR gene. The latter encodes for a receptor that specifically recognizes certain cancer antigens. This allows for a robust and potent anti-tumor response [107,120]. For example, the success of the anti-CD19 CAR T cell therapy against B cell lymphoma leading to its FDA approval is well known [121–124]. Initial testing on the first-generation CAR cells showed low efficacy and failure in immune cell activation [125–127]. For that reason, second- and third-generation CAR cells were engineered that include co-stimulatory signals that allow complete activation and expansion of T cells [128–132].

On the contrary, CAR T cell therapy has not been as successful in solid malignancies such as breast cancer mainly because of the presence of an immunosuppressive tumor microenvironment as opposed to haematological malignancies where there is none. An important factor to consider is choosing an appropriate target antigen that is highly expressed in tumor cells but not so much in healthy tissue [133]. For that reason, there are several studies that have investigated the impact of many different targets for TNBC which is one of the most aggressive types of breast cancer. For example, in vitro studies have shown that anti-AXL CAR T cells exhibit potent anti-tumor responses in TNBC cells [123,134]. Most importantly, the same anti-tumor response was also confirmed using mouse models [111,135] and Wei et al. (2018) have even reported that having a constitutively active IL-7R enhances the activity of the anti-AXL CAR T cell receptor [136]. Moreover, anti-EGFR CAR T cell receptors were reported to induce tumor lysis in vitro and perturb tumor growth in vivo [137]. There are several CAR T cells already developed and in clinical trials for TNBC treatment such as ROR1+, MUC1 and NKG2D CAR cells [138–140]. Moreover, TEM8/ANTXR1-specific CAR T cells have shown remarkable efficacy in TNBC models [141].

Overall, several other antigens have been identified so far that appear to be promising targets for the generation of such CAR T cells such as [142–144]. For example, receptor tyrosine kinases (RTKs) drive the activation of several signaling pathways such as the PI3K, MAPK or JAK/STAT pathways, leading to numerous cellular processes such as proliferation, differentiation and apoptosis [145,146]. RTKs identified in breast cancer as promising candidates for CAR cells include HER-2, EGFR and ROR1 [144,147–154]. Cell surface antigens present on tumor cells are another possible candidate for CAR cells. For instance, MUC1 upregulation was detected in the majority of breast cancer subtypes and promotes the metastatic dissemination of tumor cells through the activation of ERK1/2 and NF-kB pathways [143–145]. Furthermore, mesothelin overexpression has also been identified in aggressive breast cancer subtypes and is often associated with poor prognosis and resistance to chemotherapy treatment [143,146,147]. Currently, there are several clinical trials undergoing that investigate different targets for CAR T cells in breast cancer (Table 2—from clinicaltrials.gov). Figure 3 demonstrates the different types of ACT therapy.

**Table 2.** Clinical trials for CAR T cell therapy in breast cancer.

| Target | Breast Cancer Subtype | Phase Participants | Participants |
|---|---|---|---|
| HER2, GD2 and CD44v6 | Stage III, IV or relapsed | Not specified | 100 |
| Mesothelin | HER-2 negative | Phase I | 186 |
| ROR1 | ROR1+ relapsed or refractory TNBC | Phase I | 54 |
| cMET | Metastatic refractory cancer or operable TNBC | Not specified | 6 |
| EGFR/B7H3 | TNBC | Not specified | 30 |
| MUC1 | TN invasive | Phase I/II | 20 |
| GD2 | GD2-positive | Phase I | 94 |

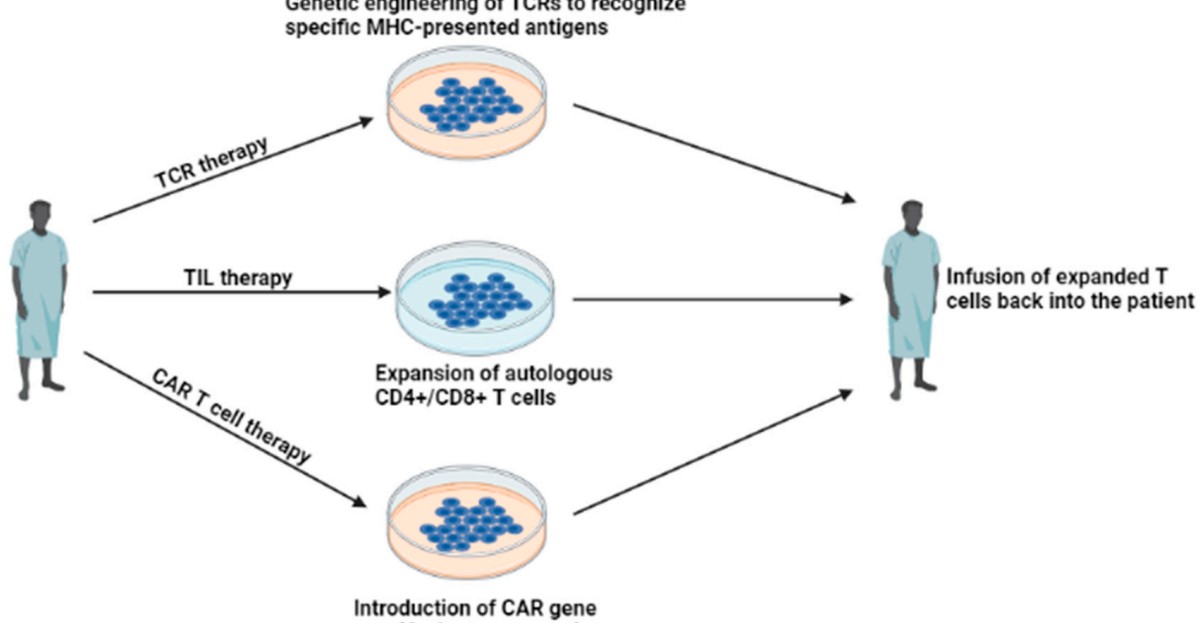

**Figure 3.** Adoptive Cellular Therapy. Illustrates the three types of ACT that exist. TIL therapy involves the ex vivo expansion of T lymphocytes. TCR therapy is the genetic modification of TCRs to specifically recognize cancer antigens. CAR T cell therapy involves the introduction of a CAR gene specific for an antigen present in a patient's tumor (created using Biorender.com, URL accessed on 16 May 2023).

Table 2 lists some of the clinical studies currently underway that investigate the use of CAR T cells in breast cancer subtypes. The different targets, trial phase and number of participants for each study are also displayed.

## 7. Limitations

### 7.1. Immune Checkpoint Inhibitors (ICIs)

Despite the worldwide implementation of ICIs in clinical practice, there are multiple challenges which must be taken into consideration. Firstly, predictors of response are considered the most important challenge among physicians, especially in the case of non-small-cell lung cancer (NSCLC), since only 30–40% of patients seem to benefit from ICIs [148,149]. Up to date, there are gaps in clinical evidence for concurrent administration of steroids and ICIs. Retrospective studies showed that patients taking steroids had a lower progression-free survival or overall survival [150–154] but were not negatively affected while taking steroids for managing side effects [155]. It remains unclear if steroids reduce the efficacy of ICIs or if there is an eventual difference driven by a more aggressive form of disease. ICI-induced adverse effects, also known as immune-related adverse effects (irAEs), are more prevalent in organs such as the skin, lungs, liver, kidneys, gastrointestinal and nervous system. ICIs have been associated with gastrointestinal and hepatic adverse

events, with colitis being the most common side effect. According to comprehensive systematic review by Wang et al., the majority of deaths were attributed to colitis (70%) with anti-CTLA-4 therapy and pneumonitis and hepatitis (22%) with anti-PD-1/PD-L1-therapy. With combination PD-1/CTLA-4 therapy, deaths were frequently due to colitis (37%) and myocarditis (25%) [156]. The median time of onset of neurological adverse events (nAEs) was 6 weeks after ICIs treatment initiation. The incidence of severe nAEs (grade 3 and 4) with anti-CTLA4 therapy was slightly higher (0.7%) than with anti-PD1 therapy (0.4%) [157]. The incidence of ICI-induced liver dysfunction is much lower compared to diarrhoea and is reported in about 1–6% of patients, mostly at grades 1 and 2 [158,159]. The greater incidence and severity of irAEs with anti-CTLA-4 therapy compared to ani-PD-1/PD-L1 therapies reflects the potentially more dominant role of CTLA-4 compared to PD-1 as a negative T-cell regulator. It is worth mentioning that irAEs of combination ICIs are more severe than those of monotherapy [160]. Combination ICIs have been associated with endocrinopathies such as thyroid disorders and hypophysitis, which can be life-threatening if not recognized early [161].

To tackle the aforementioned challenges, microsatellite instability-high (MSI-H) or deficient mismatch repair (dMMR) could be implemented, as it correlates with stronger responses with ICIs. In fact, it has already found application in as many as 24 cancer types with the highest percentage of MSI-H in endometrial cancer (17%), gastric adenocarcinoma (9%), small intestinal malignancies (8%), colorectal adenocarcinoma (6%) and lower for breast carcinoma (0.6%) [162]. The management of grade 1–2 hepatic dysfunction generally involves close monitoring with liver tests to avoid grade 3–4 liver toxicity. The management of grade 3–4 hepatic dysfunction requires high dose intravenous glucocorticoids for 24–48 h, followed by an oral steroid over at least a period of 30 days [159]. Regarding neurological AEs, no standard treatment has yet been defined. Clinical improvements have only been reported after the ICI treatment discontinuation [160]. On the contrary, endocrinopathies are reversible and can have a small impact on the patient's quality of life when managed properly. Thus, treatment discontinuation is not necessary on the onset of hypophysitis and can be managed through the replacement of pituitary hormones, which is safe and common in clinical practice [161].

### 7.2. Breast Cancer Vaccines

BC vaccines present with good tolerance but without significant clinical benefit. Particularly, the Theratope® (STn) vaccine for metastatic BC and the NeuVax™ [Nelipepimut-S (NPS), or E75] vaccine for adjuvant BC both failed to display clinical benefit during phase 3 studies [163,164]. According to the E75 Phase 2 trial, the effect of BC vaccines seems to fade over time, leading to increased recurrences. The short-lasting effect may be attributed to the following factors: poor vaccine formulations, immune tolerance to specific antigens and immune suppressive microenvironment [165].

Allogeneic vaccines offer a promising solution when the immunosuppressive microenvironment becomes an insurmountable barrier for the immune system to overcome. Taking into account the up-to-date limited clinical benefit of BC vaccines, allogeneic vaccines could be implemented in clinical practices as adjuvant treatment [166]. It should be noted though that the extensive intratumoral heterogeneity poses a challenge for the development of allogeneic vaccines [167]. The clinical benefit may also be amplified through the co-administration of multiple antigens alongside BC vaccines. Through the years, there has been progress made with respect to delivery methods, with the development of lipid nanoparticles to genetically modified viruses; however, the extent to which the patient's immune systems can respond remains unknown [168].

### 7.3. Cytokine Therapies

Just like with every treatment, cytokine therapy has its own limitation—not particularly specific for breast cancer. The major challenges presented here originate from the core principles of cytokines. First of all, being pleiotropic makes their targeting not specific

to the TME; thus, off-target effects often occur. Additionally, due to the fact that different cytokines can regulate the same pathways, blocking one of them may not be enough as the others can compensate for that loss. Finally, due to their ability to regulate the immune system of the host, a blockade of any cytokines can lead to impairment of the immune system and can lead to autoimmunity or tissue damage [169]. The introduction of point mutations or fusion with their receptor counterparts is a way of overcoming the broad specificity of cytokines [170]. For example, IL-2 can activate both T cells and NKs but regulatory T cells (Tregs) as well. The latter lead to immunosuppression whereas the former to activation of the immune system [171]. Several studies have used site-directed mutagenesis to reduce the affinity of IL-2 for the receptor on Tregs. Overall, these mutants have shown comparable activity to their wild-type forms and enhanced efficacy in in vivo settings [172–174].

To reduce the off-target effects and adverse reactions reported with cytokine therapy, studies have looked into the fusion of these cytokines with targeting antibody that directs the cytokine to the TME [170]. In cancer, these antibodies often target oncogenic antigens which are usually overexpressed such as HER-2 in HER-2+ breast cancer [175]. For example, F16-IL2 consists of IL-2 fused with an antibody against the large isoform of tenascin c which is undetectable in healthy tissues but is overexpressed within the TME [176]. In vivo models of breast cancer have shown that F16-IL2 had a superior efficacy to IL-2 alone. Furthermore, when combined with either doxorubicin or paclitaxel, survival rates and overall response drastically increased, respectively [177]. Di Trani et al. (2022) mention some of the clinical trials underway that investigate the use of these antibody-cytokine complexes [170].

Cytokines often are presented with a short half-life which limits their activity and are therefore given in multiple injections. Consequently, this leads to systemic toxicity [162]. Protein PEGylation is widely used to extend a protein's half-life as it shields the protein from immune cells and prevents proteolysis [178]. Several PEGylated cytokines (e.g., IL-2 or IL-10) are currently under investigation in clinical trials for patients with breast cancer amongst other solid malignancies (NCT02983045, NCT02009449). Some of these PEGylated cytokines have even been combined with ICIs and have demonstrated higher efficacy and tumor control compared to their monotherapy counterparts [179].

### 7.4. Adoptive Cellular Therapies (ACTs)

Each type of ACT from TIL to CAR T cell therapy has its own disadvantages as well. For example, TIL therapy is limited by the fact that we use the patient's own T cells. Some patients have non-functional T cells or even no T cells at all as they have what is known as a "cold" tumor (i.e., not immunogenic) [180,181] and therefore must be further modified to be applied in those patients. For example, they must either be co-cultured with IL-2 to further expand and activate the cells or select T cells that are tumor-specific [182]. Furthermore, there are cases where these TILs do not exhibit potent anti-tumor responses and must therefore be combined with additional therapies as well. For example, several clinical trials are combining TILs with ICIs in tumors other than breast cancer to suppress the negative co-stimulation of T cells and promote their activation instead (NCT02621021; NCT03645928; NCT03215810).

We often tend to compare CAR cells with TCR-modified cells as one can be the solution for a challenge posed by the other. For example, TCR-modified T cells target specific antigens; however, the fact that they rely on MHC-presented antigens limits their application. This can be overcome by CAR cells which target a broader range of cancer-associated antigens independent of MHC presentation [183]. On the other hand, antigen recognition by CAR cells is limited to surface antigens whereas TCR-modified cells can recognize both surface and intracellular proteins given that they are presented by HLA molecules [183–185]. One major challenge posed by solid malignancies such as breast cancer compared to haematological cancers is the presence of an immunosuppressive TME which can dampen the activity of these therapies [186–188]. This is one of the main reasons why ACT has reported remarkable success in blood rather than solid cancers and why

it is often combined with ICIs in solid malignancies as aforementioned. Finally, CAR T cell therapy in particular has often be associated with a severe side-effect profile due to inappropriate activation of T cells leading to excessive cytokine release, also known as cytokine release syndrome (CRS) [189–191]. Moreover, a wide range of neurotoxic side effects are also observed with CAR cells which may be associated with cytokines surpassing the blood-brain barrier and gaining entry to the central nervous system (CNS) [192,193].

## 8. Conclusions

Immunotherapy has revolutionized the treatment of breast cancer, significantly improving the curative effect when added to standard therapies. Despite challenges, immunotherapy remains a promising therapeutic strategy and ongoing trials will confirm its clinical benefit in combination with conventional therapies. Further knowledge about the immunosuppressive tumor microenvironment could aid the selection of patients who are most likely to respond to the aforementioned regimens.

**Author Contributions:** Conceptualization, M.V. and S.P.; methodology, M.V. and S.P.; resources, M.V. and S.P.; writing—original draft preparation, M.V. and S.P.; writing—review and editing, M.V., S.P. and N.P.N.; visualization, M.V., S.P. and N.P.N. All authors have read and agreed to the published version of the manuscript.

**Funding:** This research received no external funding.

**Institutional Review Board Statement:** Not applicable.

**Informed Consent Statement:** Not applicable.

**Data Availability Statement:** Not applicable.

**Acknowledgments:** Figures 1–3 were created using https://biorender.com. URL accessed on 16 May 2023.

**Conflicts of Interest:** The authors declare no conflict of interest.

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
