# Peer review of "Current Advancements and Future Perspectives of Immunotherapy in Breast Cancer Treatment"

_2673-5601, doi:10.3390/immuno3020013_

Round 1

Reviewer 1 Report

PAGE 2 line 51, define BC, unify if it refers to BC or breast cancer in the text

Page 3 figure caption define L.H.S., R.H.S and explain the figure

Page 2 line 82 delete the abbreviation (ICIs) and mention it up to the subtitle of page 4 line 151

Some abbreviations are defined but most are not, take care of this in the text

Page 7 CCL2, CCR2 line 292, PG line 294; define

On pages 5 (table 1) and 8 (line 330) TNBC is mentioned without defining it, and it is not defined until page 9 line 375

Author Response

Thank you for your review. According to your comments we made the corrections below:

Page 2 line 50 refers to breast cancer (BC).

Page 3 figure 1 is updated and includes an extensive caption.

Abbreviation ICIs deleted from page 2 and is first mentioned in page 5 line 184. 

Page 9 line 357 defines CCL2, CCR2.

Page 9 line 360 defines PG.

Page 2 line 58 defines TNBC.  

Overall, we included abbreviations which were not defined.

There are additions in the introduction & main body and changes in figure 1 and 2 based on the comments of the 2nd reviewer. Also, we included table 2 and figure 3 in the manuscript, which were accidentally not included by the managing editor, since we had attached them separately.

Please let me know if you have further comments.

Reviewer 2 Report

The authors have provided a fairly comprehensive overview of the current status of immunotherapy for breast cancer. Even though the review is supposed to describe it for breast cancer in general, it seems to be heavily skewed in favor of HER2+ related developments with little attention to TNBC, which is in fact a "hotter" subtype with much more activity in the area of immunotherapy. This needs to be addressed in more depth in each category of therapies. Additionally, I have a few suggestion which need to be addressed for better flow of information

Assuming the target audience are readers who are unfamiliar with breast cancer and/or immunotherapy, it is essential to include some introduction (beyond statistics of incidence rates etc.) to breast cancer subtypes, the immune system and the tumor immunity cycle. Additionally, Fig 1 needs to be modified to include other checkpoints as well as a proper dimerization event for HER2. The figures in their current form are misleading and do not add anything to the text. Fig 3 is practically wrong since it completely omits the process of showing antigen presentation by APCs which leads to T cell activation. 

Author Response

Thank you for your review. According to your comments we made the corrections below:

We included an introduction to the immune system. We refer to the definition and phases of immunosurveillance as well as the definition of immunogenicity. We also included an introduction to breast cancer subtypes that are widely recognized and we elaborated on TNBC in each category of immunotherapies.

We modified Figure 1 which now includes other checkpoints and the HER2 dimerization process, and Figure 2 which now includes the antigen presentation process by APCs.

Lastly, we included table 2 and figure 3 in the manuscript, which were accidentally not included by the managing editor, since we had attached them separately.

Please let me know if you have further comments.
